# Development of a prostate cancer biochemical recurrence risk signature using machine learning and motor protein-related genes

Weixing Wang[1]*, Guohai Xie[2], Zhonggao Wu[1]

1 Department of Urology, Ningbo Haishu People's hospital, Ningbo, Zhejiang, China, 2 Department of Urology,The First Affiliated Hospital of Ningbo University, Ningbo, Zhejiang, China

* 13486485400@163.com

## Abstract

### Background

Motor proteins play significant roles in cancer progression, but their involvement in biochemical recurrence (BCR) of prostate cancer remains unclear. The objective of the study is to develop a prognostic indicator for BCR using machine learning (ML) and motor protein-related genes (MPRGs).

### Methods

The prognosis relevance of the MPRGs in prostate cancer was analyzed by univariate Cox regression. Feature selection and model construction were performed using combinations of multiple machine learning algorithms. Model performance was assessed using receiver operating characteristic curve and C-index. Patients were stratified into high- and low-risk groups based on the risk signature, and comparisons of BCR incidence, gene expression profiles, immune cell infiltration patterns, and drug sensitivity were conducted between these groups. The gene expression of MPRGs were validated in vitro.

### Results

Among 120 MPRGs, 17 were differentially expressed, of which 8 were significantly associated with BCR. A novel risk scoring system using a StepCox[forward] + Ridge model based on these 8 MPRGs effectively stratified patients into two different risk groups, and patients with high riskscores had significantly higher BCR rates than those with lower riskscores. Enrichment analysis revealed upregulation of inflammation response, EMT, hypoxia, and estrogen response pathways in the high-risk category, while mitotic spindle, G2M checkpoint, and E2F targets were downregulated. The MPRG-derived risk score correlated positively with M2 macrophage infiltration and ngatively correlated with CD4 T cells and mast cells, and the high-risk category

**Data availability statement:** The RNA expression profiles and corresponding clinical information used in this study are available in the TCGA (https://portal.gdc.cancer.gov), cBioPortal for Cancer Genomics (https://www.cbioportal.org/) and GEO(https://www.ncbi.nlm.nih.gov/geo/query/acc.cgi?acc=GSE70769) databases. All relevant data are within the manuscript and its Supporting information files.

**Funding:** The author(s) received no specific funding for this work.

**Competing interests:** The authors have declared that no competing interests exist.

showed higher sensitivity to drugs like cisplatin and bicalutamide. The final nomogram based on MPRGs-derived signature and T stage provided an excellent tool for predicting BCR. In vitro experiments further validated that the expression trends of MPRGs in the risk signature were consistent with the bioinformatics analysis results.

## Conclusion

This study developed a novel MPRG-derived risk signature that effectively predicts BCR in prostate cancer, offering valuable insights for clinical management and personalized treatment strategies.

## Introduction

Prostate cancer (PCa) ranks as one of the most significant health issues affecting men globally. It is both the most prevalent cancer in males and the second leading cause of cancer-related mortality among them [1]. The importance of androgen receptor (AR) signaling in regulating the growth and survival of prostate cancer cells cannot be overstated [2], making androgen deprivation therapy (ADT), which targets this pathway, a cornerstone in the management of both early-stage and metastatic PCa [3]. Despite initial success, many patients face the grim reality of developing castration-resistant prostate cancer (CRPC) following prolonged ADT treatment, a condition characterized by poor prognosis and limited therapeutic options [4]. Additionally, around 25% to 40% of patients treated with radical prostatectomy or radiation therapy for localized PCa experience biochemical recurrence (BCR), thus highlighting the need for improved understanding and management of tumor recurrence [5]. To address these challenges, the identification of robust biomarkers that can predict PCa progression and recurrence is crucial. Such biomarkers would not only enhance the precision of active surveillance but also facilitate personalized treatment strategies and improve patient outcomes.

Motor proteins, including kinesins and dyneins, are essential for a range of cellular functions, including intracellular transport, cell division, and signal transduction [6]. Recent studies have increasingly highlighted the involvement of these proteins in cancer biology, suggesting that alterations in motor protein expression or function may contribute to tumor initiation, progression, and metastasis [6]. For instance, certain kinesin family members have been shown to regulate the mitotic spindle assembly and chromosome segregation, processes that are frequently disrupted in cancer cells, leading to genomic instability [7]. Similarly, dysregulation of dynein-dependent pathways has been linked to the aberrant trafficking of key signaling molecules, which can promote tumor proliferation and invasion [8]. Given the pivotal roles of motor proteins in maintaining cellular homeostasis, their aberrant expression or activity may serve as potential biomarkers for cancer diagnosis, prognosis, and therapeutic intervention [9,10]. In PCa, understanding how changes in motor protein dynamics influence disease progression and response to therapy may offer important perspectives on the development of targeted therapies and personalized medicine approaches.

This study aims to explore the correlations between motor protein-related genes (MPRGs) and PCa outcomes, focusing on their predictive value for patient prognosis. By elucidating these relationships, we hope to identify new therapeutic targets and improve the management of this complex disease.

## Materials and methods

### Data acquisition

Transcriptomic profiles and clinical annotations of PCa patients were systematically retrieved from three independent repositories. (1) TCGA-PRAD cohort (n = 321) and clinical metadata were accessed through the Genomic Data Commons (GDC) Data Portal (TCGA, https://portal.gdc.cancer.gov). (2) The MSKCC2010 cohort (n = 140) was obtained from cBioPortal for Cancer Genomics (https://www.cbioportal.org/). (3) Publicly available transcriptomic data (GSE70769) were downloaded from the Gene Expression Omnibus (GEO, https://www.ncbi.nlm.nih.gov/geo) database, containing 92 PCa specimens. To ensure cross-platform compatibility, all raw RNA-seq data underwent standardized processing: FPKM quantification was performed using STAR aligner (v2.7.10a) with GENCODE v35 reference transcriptome; Cross-cohort batch effects were mitigated via the sva::ComBat function (R v4.2.1) after log2(FPKM+1) transformation. Additionally, we retrieved 120 MPRGs from the Kyoto Encyclopedia of Genes and Genomes (KEGG, https://www.kegg.jp/) database (Supplementary information: S1 Table).

### Machine learning algorithms for risk signature construction

The prognostic signature was constructed through an integrative machine learning framework using the MIME1 package (v0.0.0.9000). Feature selection was initiated with univariate Cox regression analysis performed on the TCGA-PRAD cohort, identifying MPRGs significantly associated with BCR (p < 0.05). Subsequent model optimization employed the ML.Dev.Prog.Sig() function with standardized parameters: nodesize = 6, mode = "all". Model selection was performed based on the mean C-index derived from validation cohorts, with survival models ranked by predictive performance. Using the median risk score as the cutoff, patients were dichotomized into high- and low-risk groups. Optimal models were selected based on consistent statistical significance across all cohorts, defined by Log-Rank tests (p < 0.05 for group differences in BCR-free survival). Time-dependent area under the receiver operating characteristic curve (AUC) analysis was conducted using the cal_AUC_ml_res() function with Kaplan-Meier survival estimators, quantifying predictive accuracy for BCR at 1-, 3-, and 5-year intervals.

### Functional enrichment analysis

Limma package was used to identify differentially expressed genes (DEGs) between groups (adjusted p value < 0.05 and |log2(fold change)| > 1). Gene Ontology (GO) and KEGG pathway enrichment analyses were conducted using the clusterProfiler package [11]. Hallmark gene sets were downloaded from the Molecular Signature Database (MSigDB, https://www.gsea-msigdb.org/), and were used to conduct gene set enrichment analysis.

### Immune infiltration analysis

The IOBR package was used for evaluation of immune infiltration [12], employing the CIBERSORT algorithm to compute the infiltration levels of 22 immune cell types in the TCGA-PRAD cohort.

### Drug sensitivity analysis

Drug sensitivity was analyzed using the pRRophetic package [13] for 45 chemotherapeutic drugs. Differences in drug sensitivity between groups were evaluated using the Wilcoxon rank-sum test. Pearson correlation analysis was conducted to assess the correlations between gene expression and drug response.

## Development of a nomogram

The prognosis relevance of the riskscore and other clinicopathological features was analyzed using univariate and multivariate Cox regression analyses. A nomogram model was built using the independent prognostic factors ($p < 0.05$) of the TCGA-PRAD cohort with the rms package. The performance in BCR prediction in PCa was analyzed by calibration curves, which compare the predicted probabilities of BCR-free survival at 1-, 3-, and 5-year intervals with the observed probabilities derived from Kaplan-Meier estimates. The calibration curves were generated using the 'rms' package in R, with bootstrapping (n = 1000 resamples) to account for overfitting. Additionally, decision curves, and ROC curve analyses were also performed. The AUC values for 1 -, 3 -, and 5 – year BCR of the nomogram for the MSKCC2010 and GSE70769 cohorts were calculated to validate the robustness of the nomogram performance.

## Cell lines and cell culture

Human PCa cell lines DU145 (catalog no. CL-0075) and LNCaP (catalog no. CL-0143) were acquired from Procell Life Science & Technology Co., Ltd. (Wuhan, China), while the benign prostate hyperplasia epithelial cell line BPH-1 (catalog no. AC339850) was sourced from Shanghai Zeye Biotechnology Co., Ltd. (Shanghai, China). DU145 cells were grown in MEM medium supplemented with non-essential amino acids, 10% fetal bovine serum (FBS), and 1% penicillin/streptomycin (P/S). LNCaP cells were propagated in RPMI-1640 medium containing 10% FBS and 1% P/S. BPH-1 cells were maintained in high-glucose DMEM (90%) with 10% FBS and 1% P/S. All cell lines were incubated at 37°C in a humidified atmosphere with 5% $CO_2$.

## RNA extraction and quantitative analysis

Total RNA was isolated from the cell lines using TRIzol Reagent (Invitrogen, USA). Complementary DNA (cDNA) synthesis was performed with PrimeScript RT Master Mix (Takara, USA), followed by quantitative real-time PCR (qRT-PCR) using SYBR qPCR Master Mix (Vazyme, China). Gene expression levels were normalized to GAPDH as the reference gene, with relative quantification calculated via the $2^{-\Delta\Delta Ct}$ method. Primer sequences are provided in Supplementary information: S2 Table.

## Statistical analysis

All statistical analyses were performed using R (v4.5.0). Data were presented as mean±standard deviation and were compared between groups through one-way analysis of variance. Statistical significance was defined as a two-sided P-value $< 0.05$.

## Results

### Expression and mutation characteristics of MPRGs in prostate cancer

First, we removed batch effects between the three cohorts (Figs 1A, B). Differential expression analysis revealed that among 120 MPRGs, 17 were differentially expressed, with 10 significantly upregulated and 7 significantly downregulated in prostate cancer (Fig 1C). The heatmap of these differentially expressed MPRGs is shown in Fig 1D. Univariate Cox regression analysis identified 8 MPRGs significantly associated with BCR in prostate cancer, including kinesin family member C2 (KIFC2), kinesin family member C1 (KIFC1), kinesin family member 18B (KIF18B), kinesin family 14 (KIF14), kinesin family 11 (KIF11), centromere protein E (CENPE), actin alpha cardiac muscle 1 (ACTC1), and kinesin family 15 (KIF15) (Fig 1E). As shown in Fig 1F, 16.67% (20/120) of MPRGs harbored somatic mutations, with the top three most frequently mutated genes being MYH6 (9%), KIF16B (7%), and KIF11 (6%).

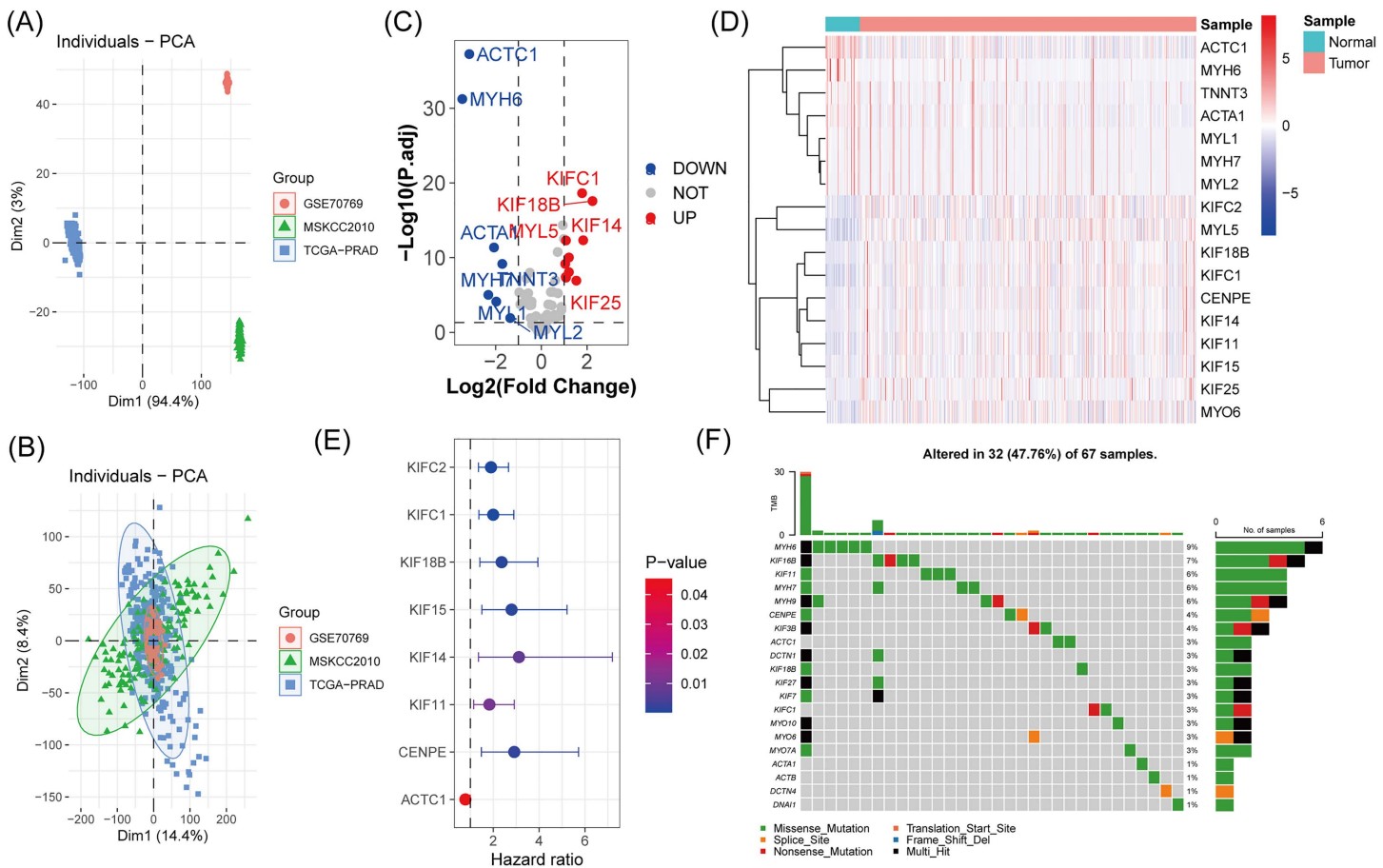

**Fig 1. Expression and mutation characteristics of MPRGs in PCa.** PCA plot of the three cohorts (A) before batch correction and (B) after batch correction. C: Volcano plot of differential expression analysis for MPRGs in PCa. D: Heatmap of differentially expressed MPRGs. E: Univariate Cox regression analysis identifying MPRGs significantly associated with BCR. F: Somatic mutation landscape of MPRGs in PCa.

## Feature selection and risk signature construction using machine learning

Using the TCGA-PRAD cohort, we identified eight MPRGs significantly associated with prostate cancer BCR. A comprehensive evaluation of 117 machine learning algorithm combinations and their C-index performance across multiple cohorts (Fig 2A) led to the selection of the StepCoxforward+Ridge model as the optimal approach. Stratifying patients by the median risk score derived from this model, we observed significantly elevated BCR risks in high-risk groups compared to low-risk groups, with hazard ratios (HR) of 4.79 (p<0.001) in TCGA-PRAD, 2.3 (p=0.012) in MSKCC2010, and 2.03 (p=0.016) in GSE70769, as demonstrated by Kaplan-Meier survival analysis (Fig 2B). Time-dependent ROC curve evaluations revealed robust predictive performance. The area under the curve (AUC) values for 1-year BCR prediction were 0.696 (TCGA-PRAD), 0.835 (MSKCC2010), and 0.749 (GSE70769); corresponding 3-year AUC values were 0.721, 0.689, and 0.710, while 5-year AUC values maintained predictive utility at 0.700, 0.664, and 0.649 across cohorts (Fig 2C).

## Association between risk score and T, N staging, and age

The heatmap of the 8 MPRGs showed that, except for ACTC1, the remaining genes were predominantly upregulated in the high-risk category and downregulated in the low-risk category (Fig 3A). PCa patients who were diagnosed at the T3

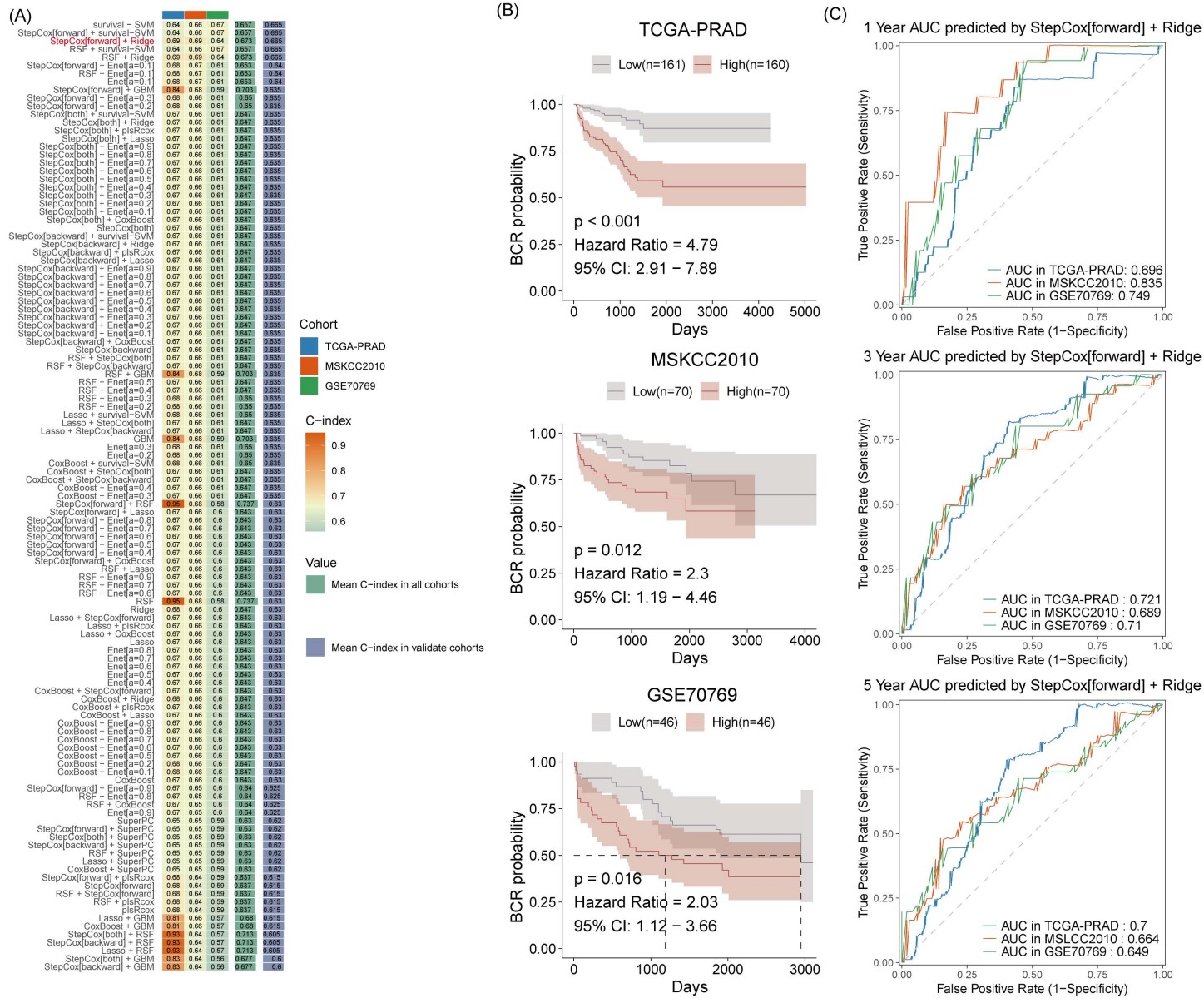

**Fig 2. Performance of the MPRG-based StepCox[forward] + Ridge model in predicting BCR.** A: C-index values for selected machine learning algorithms. B: Kaplan-Meier survival curve analysis between High and Low groups in the three cohorts. C: Receiver operating characteristic curve analysis of risk scores for predicting 1 -, 3 -, and 5 – year BCR in the three cohorts.

stage presented significantly elevated riskscores than those diagnosed at the T2 stage (Fig 3B). Similarly, patients diagnosed at the N1 stage exhibited significantly higher riskscores compared to those diagnosed at the N0 stage (Fig 3C). Age was positively correlated with riskscores (r = 0.18, p = 0.0011, Fig 3D), suggesting that older patients tend to have higher riskscores than younger individuals (Fig 3E). These findings suggest that riskscores were closely correlated with tumor size, nodal invasion and age.

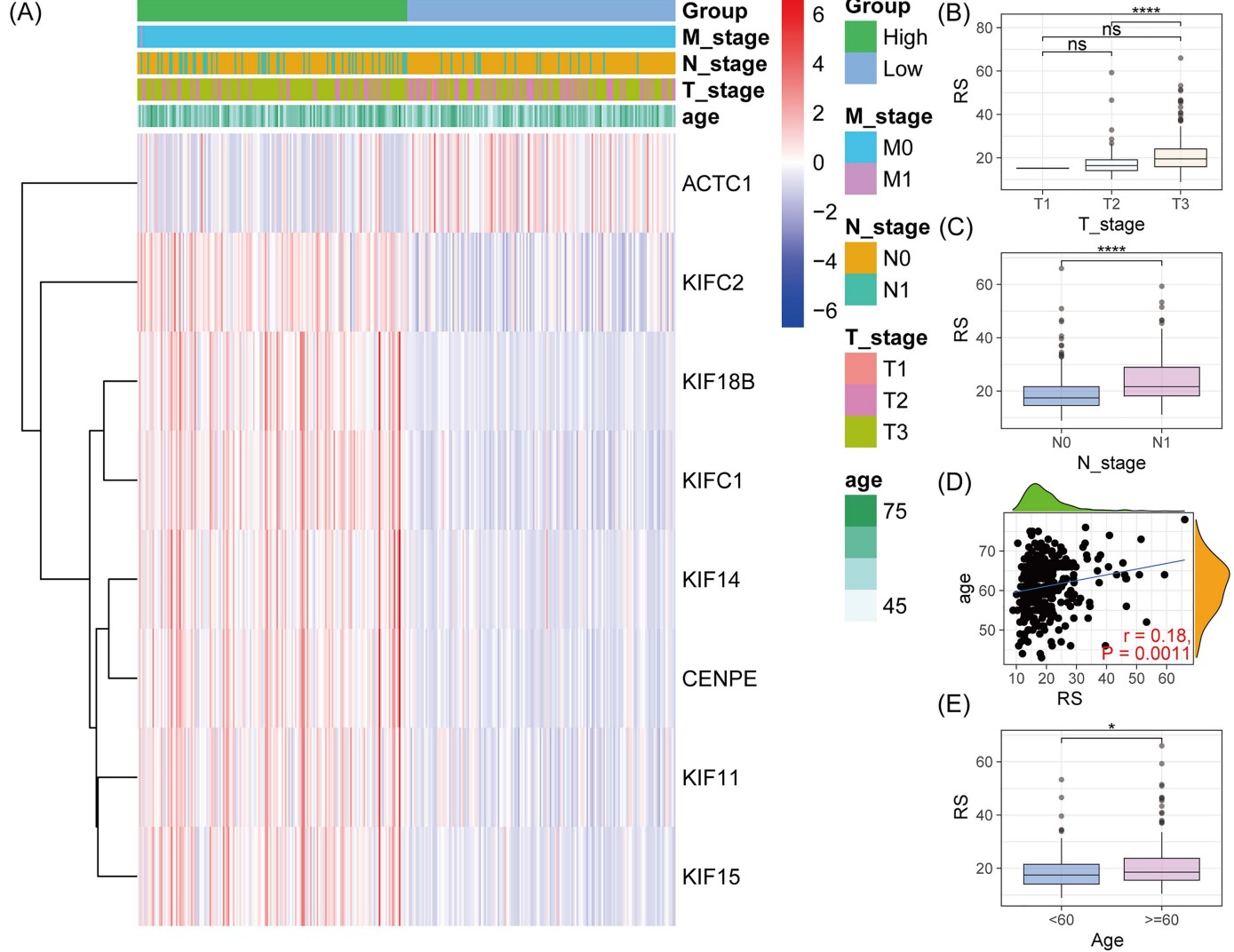

**Fig 3. Association between clinical pathological features and risk score.** (A) Heatmap of the 8 MPRGs in different risk groups. (B) Comparison of risk scores between T2 and T3 stages. (C) Comparison of risk scores between N0 and N1 stages. (D) Scatter plot and (E) box plot showing the correlation between age and risk score.

## Gene expression related to risk score

To understand the mechanisms underlying the risk score's predictive power, we performed differential expression analysis, which identified 351 DEGs between two risk groups (Fig 4A). It was revealed that multiple pathways such as the inflammatory response, epithelial-mesenchymal transition (EMT), hypoxia, and estrogen response were significantly activated in patients with higher riskscores, whereas pathways like mitotic spindle, G2M checkpoint, and E2F targets were suppressed (Fig 4B). Additionally, DEGs between groups were found to be enriched in processes like nuclear division and chromatid segregation (Fig 4C), as well as KEGG pathways related to the cytoskeleton in muscle cells, motor proteins, cell cycle, calcium signaling pathway, salivary secretion, and protein digestion and absorption (Fig 4D).

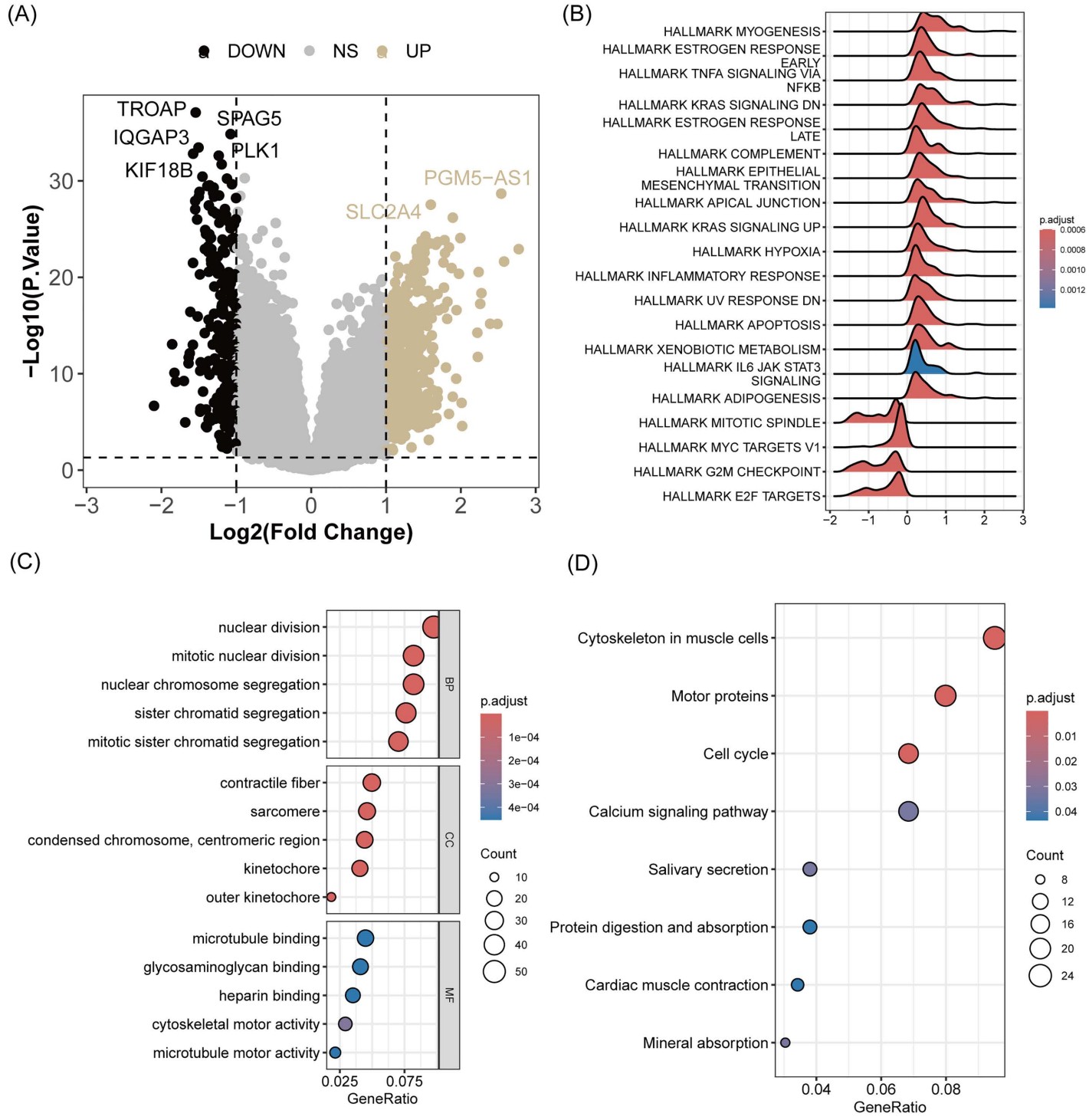

**Fig 4. Gene expression changes related to risk score.** (A) Volcano plot of DEGs between high- and low-risk groups. (B) Gene Set Enrichment Analysis (GSEA) of hallmark gene sets. (C) GO enrichment analysis and (D) KEGG pathway enrichment analysis of DEGs.

## Association between risk score and M2 macrophage infiltration

To evaluate whether the MPRG-derived risk score indicates the tumor immune microenvironment (TIME), the infiltration levels of 22 immune cell types were calculated and analyzed. As a result, the levels of M2 macrophages and Tregs were dramatically elevated in individuals with higher riskscores compared to those with lower riskscores, whereas low-risk groups demonstrated greater CD4 + T cell and mast cell infiltration (Fig 5A). Furthermore, risk score demonstrated a significant positive correlation with M2 macrophage infiltration and inverse associations with mast cell and CD4 + T cell infiltration (Fig 5B). These data suggest that the MPRG-derived riskscore correlated with increased immunosuppressive M2 macrophage dominance and reduced cytotoxic CD4 + T cell presence, potentially driving a more immunosuppressive tumor microenvironment.

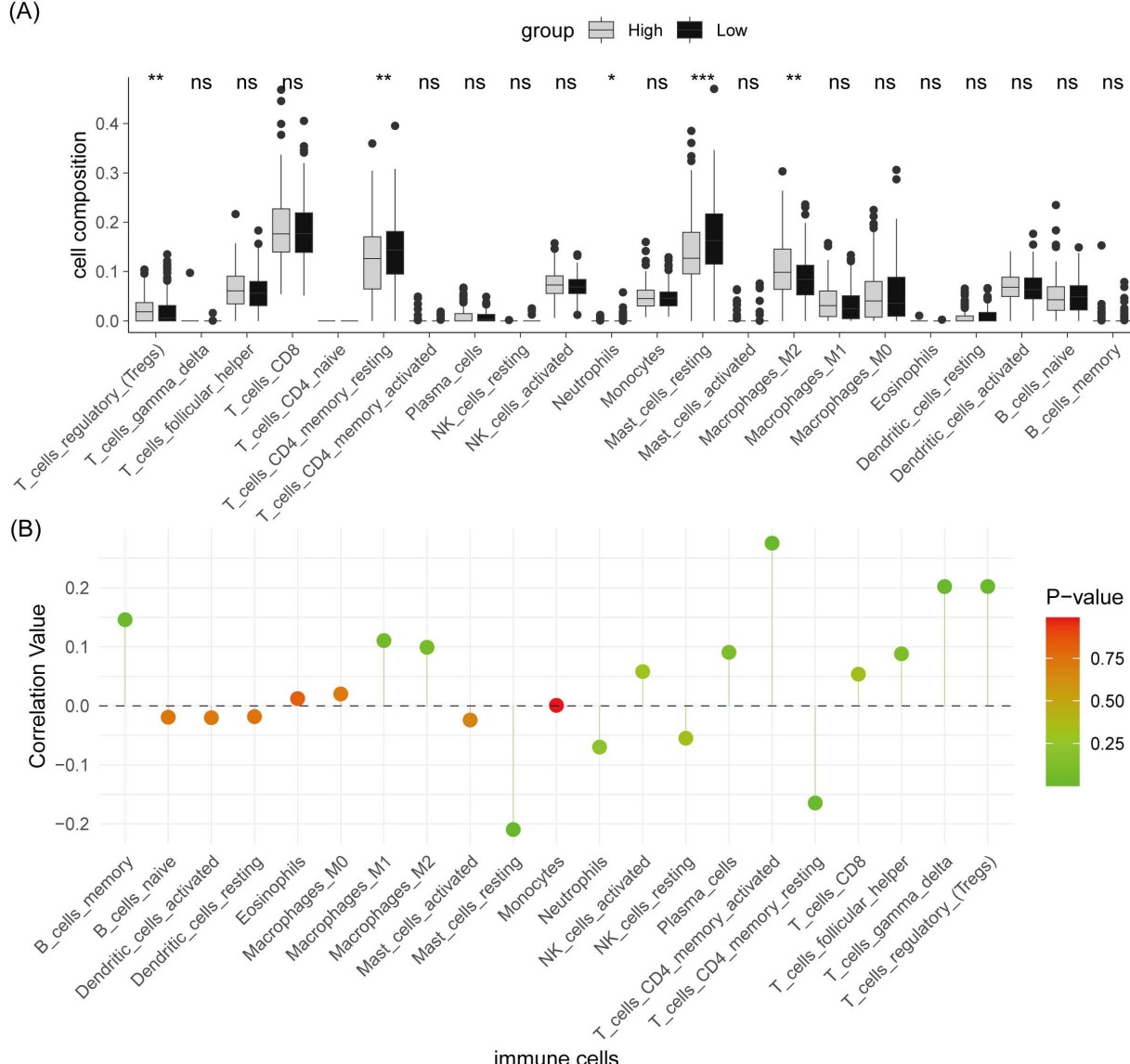

**Fig 5. Correlation between risk score and immune infiltration.** (A) Comparison of immune cell infiltration proportions between high- and low-risk groups. (B) Pearson correlation analysis between risk score and immune cell infiltration.

## Association between risk score and drug response

We further evaluated the differences in drug sensitivity across two risk categories. It was demonstrated that the low-risk category exhibited higher sensitivity to Vinorelbine, Vinblastine, Shikonin, Salubrinal, Obatoclax.Mesylate, Methotrexate, Lenalidomide, Cyclopamine. Meanwhile, the low-risk category also presented lower sensitivity to Temsirolimus, Midostaurin, Lapatinib, Imatinib, Erlotinib, Embelin, Dasatinib, Cisplatin, Bortezomib, Bicalutamide, Bexarotene and Axitinib (Fig 6A). Additionally, our data showed that MPRGs expression was closely associated with drug sensitivity, with ACTC1 showing an inverse relationship compared to other MPRGs (Fig 6B).

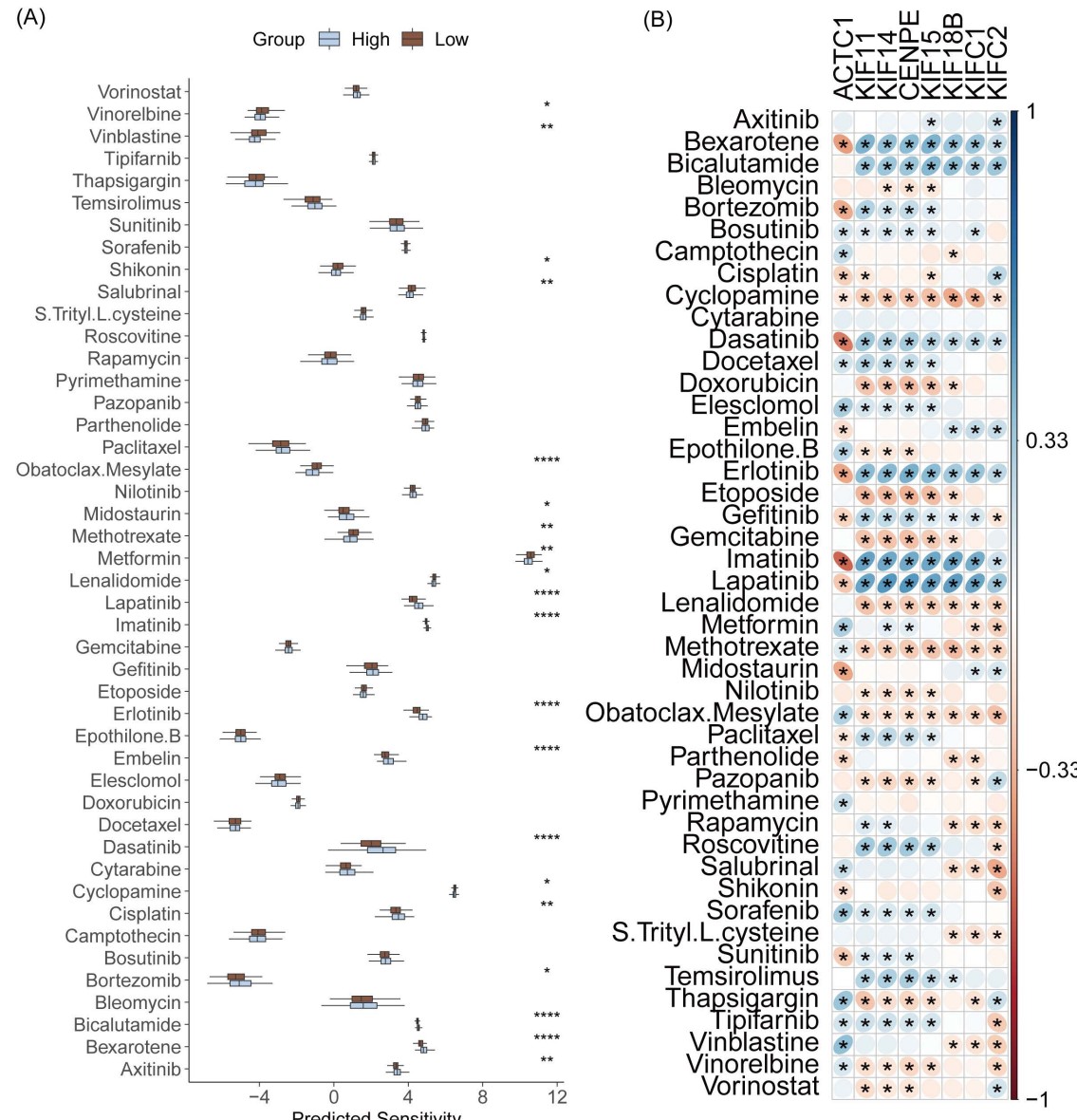

**Fig 6. Association between risk score and drug sensitivity.** (A) Comparison of drug sensitivity between high- and low-risk groups. (B) Pearson correlation analysis between MPRG expression and drug sensitivity.

## Construction of nomogram using MPRG-derived risk score

Next, we identified factors associated with BCR of PCa. As a result, riskscore, T stage, N stage, and PSA were significantly associated with the BCR in PCa. Furthermore, riskscore, T stage, and PSA were identified as independent prognostic indicators for BCR (Table 1). Therefore, we generated a nomogram model using these variables (Fig 7A). Calibration curves demonstrated strong agreement between the nomogram-predicted probabilities of BCR-free survival and the actual observed probabilities across the TCGA-PRAD cohort, with the ideal calibration line (45-degree diagonal) closely aligning with the plotted data points, indicating minimal systematic errors (Fig 7B). Additionally, the nomogram had a higher standardized net benefit compared to other factors, as shown in the decision curves (Fig 7C). Furthermore, in the TCGA-PRAD cohort, the accuracy in 1-, 3-, and 5-year BCR prediction of the nomogram was 0.724, 0.76 and 0.741, respectively (Fig 7D). In the GSE70769 cohort, corresponding AUC values reached 0.802, 0.728, and 0.694 (Fig 7E), respectively.

## Validation of the mRNA expression levels of MPRGs in PCa cells

To further validate the mRNA expression profiles of these eight genes, we quantified their levels in prostate cell lines using RT-qPCR. The results revealed that, compared to the BPH-1 cell line, the mRNA levels of KIF14, KIF11, CENPE, KIF15, KIFC2, KIF18B, and KIFC1 were significantly upregulated in PCa cell lines (LNCaP and DU145), whereas ACTC1 expression was markedly downregulated. These findings align with the observations from the TCGA-PRAD cohort. Furthermore, relative to the non-castration-resistant prostate cancer cell line (LNCaP), the expression of KIF14, CENPE, KIF15, KIF18B, and KIFC1 was significantly elevated in the castration-resistant PCa cell line (DU145) (Fig 8). Given that castration-resistant PCa cell lines exhibit greater malignancy than their androgen-sensitive counterparts, these data suggest that elevated expression of these MPRGs positively correlates with the degree of prostate cancer aggressiveness, thereby corroborating the reliability of our prior bioinformatics analyses.

## Discussion

Prostate cancer is the most common malignant tumor in the urinary system and one of the most prevalent cancers in men [14]. Exploring new biomarkers for tumor development, particularly those for BCR, can aid in early classification of PCa patients and provide appropriate treatment options. Motor proteins, as well as microtubules (MTs) and other tubulin- and actin-based structures, are essential for the proliferation and invasive capabilities of cancer cells [6]. Recent studies have shown that KIFC2 promotes prostate cancer progression and chemotherapy resistance by mediating NF-κB p65 expression and nuclear translocation [15]. Silencing and inhibition of KIFC1 have been reported to reduce PCa cell viability through the induction of multipolar mitosis and apoptosis, suggesting that inhibition of KIFC1 by AZ82 might be a therapeutic strategy for controlling PCa cell proliferation [16]. Additionally, multiple studies have highlighted the potential prognostic value of MPRGs in BCR of prostate cancer [9,17].

**Table 1. The results of univariate and multivariate cox regression analysis.**

| Characteristics | Univariate Cox | | | | Multivariate Cox | | | |
|---|---|---|---|---|---|---|---|---|
| | HR | lower.95 | upper.95 | p-value | HR | lower.95 | upper.95 | p-value |
| riskscore | 1.00 | 1.00 | 1.10 | 0.0001 | 1.03 | 1.01 | 1.06 | 0.0088 |
| age | 1.00 | 0.99 | 1.10 | 0.2000 | 1.00 | 0.97 | 1.04 | 0.8499 |
| T_stage | 4.90 | 2.10 | 11.00 | 0.0002 | 3.66 | 1.53 | 8.76 | 0.0036 |
| N_stage | 2.00 | 1.20 | 3.60 | 0.0110 | 1.20 | 0.67 | 2.15 | 0.5310 |
| PSA | 1.10 | 1.00 | 1.10 | 0.0001 | 1.07 | 1.04 | 1.10 | 0.0001 |
| treatment | 0.87 | 0.59 | 1.30 | 0.4900 | 0.98 | 0.67 | 1.42 | 0.9069 |

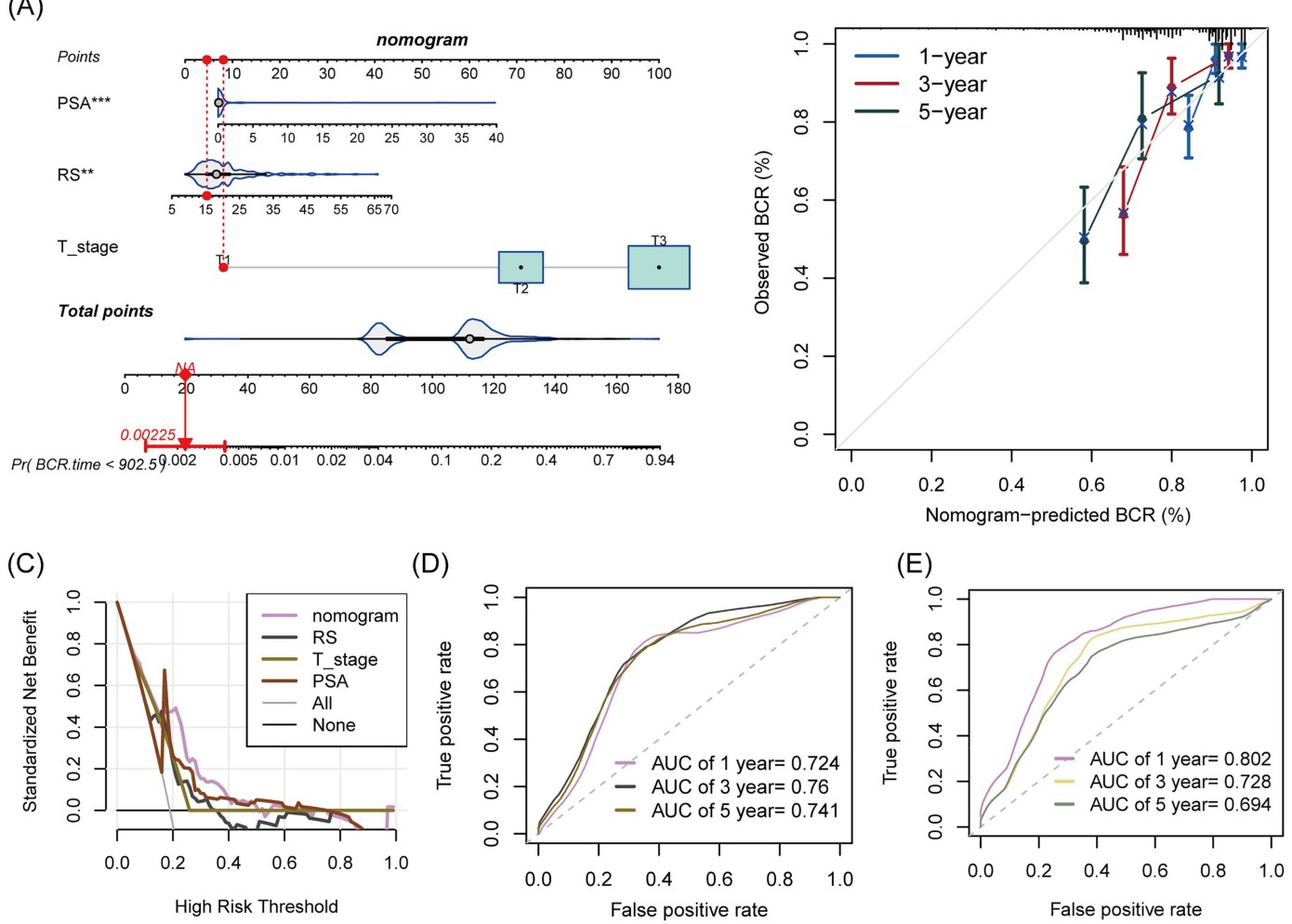

**Fig 7. Construction of nomogram using MPRG-derived risk score to predict 1-, 3-, and 5-year BCR in prostate cancer.** (A) Nomogram constructed using the independent risk factor. (B) Calibration curve, (C) Decision curve, and (D) ROC curve analysis for the nomogram in the TCGA-PRAD cohort. (E) ROC curve analysis for the nomogram in the GSE70769 cohort.

In this study, we constructed an 8-MPRG-derived risk signature using machine learning algorithms and validated its utility as a prognostic indicator for BCR in prostate cancer. While previous studies have identified novel signatures for predicting outcomes in prostate cancer patients using the TCGA database, results from a single database are not always convincing [18,19]. Our work integrated multiple datasets to generate the risk signature and employed various methods to characterize the riskscore.

The 8-gene MPRG-derived risk signature incorporates genes whose roles in prostate cancer biology and prognosis have been extensively investigated (Table 2) [9,10,15–17,20–34]. For instance, the KIFC1 inhibitor CW069 can trigger apoptosis in PCa cells and overcome docetaxel resistance [21]. Silencing and inhibiting KIFC1 also reduce PCa cell viability by inducing multipolar mitosis and apoptosis [16]. KIF18B promotes PCa cell growth, migration, and invasion while inhibiting apoptosis, possibly through activating the PI3K-AKT-mTOR signaling [24]. Docetaxel, a first-line chemotherapeutic agent for CRPC, has been shown to exhibit cross-resistance mediated by KIF14, suggesting that targeting KIF14

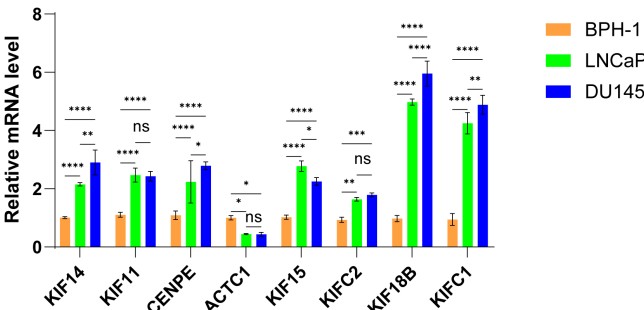

**Fig 8. Validation of the mRNA level of the 8 MPRGs in the signature.** Comparison of the relative mRNA level of the 8 MPRGs among the PCa cell lines.

**Table 2. Summary of functions and mechanism of prognostic MPRGs in PCa.**

| Symbol | Gene Name | Functions or Mechnisms in PCa |
|---|---|---|
| KIFC2 | kinesin family member C2 | regulated p65 [15] to promote progression; |
| KIFC1 | kinesin family member C1 | Associated with poor prognosis [9]; involved in multipolar mitosis and apoptosis [16]; induced docetaxel resistance [20,21]; |
| KIF18B | kinesin family member 18B | activated the PI3K-AKT-mTOR signaling pathway and regulated the proliferation and invasion [24]; associated with worse overall survival [23]; KIF18B-derived circRNA circKIF18B_003 was associated with lipid metabolism [22]; |
| KIF15 | kinesin family member 15 | Up-regulated and was associated with poorer clinical outcomes [25]; targeted by ATR-I and involved in the growth and enzalutamide resistance [25]; affected cell proliferation, tumorigenesis, migration, and cell apoptosis by regulating the PI3K/Akt pathway [26]; activated EGFR signaling pathway [27]; regulated MEK-ERK signaling [28]; stabilized AR and AR-V7 to mediate enzalutamide resistance [29]; |
| KIF14 | kinesin family member 14 | Mediated cabazitaxel-docetaxel cross-resistance by activating AKT signal [30]; correlated with tumor progression and poor prognosis [31]; |
| KIF11 | kinesin family member 11 | Predicted aggressiveness [17]; predicted bone metastasis-free survival [10]; |
| CENPE | centromere protein E | regulated by the co-binding of LSD1 and AR to its promoter [32]; increased tumor growth [32]; |
| ACTC1 | actin alpha cardiac muscle 1 | predicted bone metastasis-free survival [33]; upregulated in the skeletal muscle of men undergoing androgen deprivation therapy [34]; |

could be an effective measure to reverse docetaxel or cabazitaxel resistance or enhance their antitumor effects [30]. KIF11 has been identified as an independent predictor of bone metastasis in PCa patients and can guide clinical practice [10]. CENPE, a protein that binds to centromeres and functions as a mitotic motor, is controlled by the joint binding of LSD1 and AR to its promoter region and plays a role in the development of CRPC [32]. ACTC1 has been shown to predict metastasis-free survival in patients undergoing radical radiotherapy for prostate cancer [33], and its expression is upregulated in the skeletal muscles of men with prostate cancer receiving androgen deprivation therapy, potentially representing a compensatory response to therapy-induced muscle loss [34]. These findings suggest that the 8 prognostic MPRGs may influence the initiation and progression of PCa, although their specific mechanisms require further experimental validation.

Multiple oncogenic pathways were significantly associated with the MPRG-derived signature. Cell cycle regulation, a pivotal pathway influencing tumor growth, metastasis, and therapeutic resistance [35], was prominently linked. KRAS activation drives MAPK (RAS/RAF/MEK/ERK) and PI3K/AKT/mTOR signaling cascades, promoting proliferation, inhibiting apoptosis, and facilitating epithelial-mesenchymal transition (EMT) and metastasis [36]. EMT represents a progressive process that enhances metastatic potential, correlates with cancer stem cell expansion, and confers therapeutic resistance [37]. Hypoxia activates HIF signaling, inducing extracellular matrix remodeling, stromal transformation, and

immunosuppressive microenvironment formation [38,39]. Additionally, hypoxia triggers aberrant AR signaling activation leading to androgen deprivation therapy (ADT) resistance [40], reduces radiosensitivity, and induces chemoresistance [41,42]. Inflammatory responses drive progression and therapeutic resistance through cytotoxic factor release, immunosuppressive microenvironment formation, and metabolic reprogramming [43]. Xenobiotic metabolism modulates drug-metabolizing enzyme activity, oxidative stress, and inflammatory microenvironments, impacting cancer progression and treatment outcomes [44], with genetic variants in xenobiotic-metabolizing enzymes linked to PCa risk [45]. Spindle inhibition induces multipolar mitosis (e.g., tripolar, quadripolar), generating genomically heterogeneous daughter cells that enhance metastatic potential [46,47]. Suppression of spindle assembly enriches aneuploidy-dependent clones (e.g., TP53-deficient), conferring chemoresistance [48]. Inhibition of the G2M checkpoint pathway disrupts cell cycle arrest, allowing unrepaired DNA damage transmission and accelerating tumor evolution via chromosomal instability. Notably, elevated G2M activity correlates with immunosuppressive barrier structure (SIBS) formation, while its suppression restores CD8 + T cell function [49], potentially enhancing PD-1/PD-L1 inhibitor efficacy by reversing immunosuppression.

Currently, cancer immunotherapy has made significant strides in improving cancer outcomes [50]. PCa was characterized as an immunologically "cold" tumor, and displays an immunosuppressive microenvironment, which hinders the effectiveness of immunotherapy [51,52]. Herein, we found the correlations across riskscore and immune landscape in PCa. Tumor-associated macrophages (TAMs), primarily M2-type TAMs, are a critical component of the TIME, promoting tumor proliferation and metastasis [53]. Therefore, a higher abundance of M2-type TAMs in patients with higher riskscore may contribute to their higher rate of BCR. However, monotherapy with immune checkpoint inhibitors has shown limited efficacy in advanced metastatic prostate cancer, suggesting that a combination of targeted therapy and/or chemotherapy with immunotherapy may offer more promising treatment strategies for improving outcomes [51,54]. Individuals with lower riskscores might benefit more from chemotherapy due to their higher sensitivities to most chemotherapeutic agents.

While this in silico study provides novel insights into the prognostic value of MPRG expression signatures in prostate cancer, several important limitations must be acknowledged. First, the entirely computational nature of this investigation inherently limits biological interpretation, as we cannot confirm whether observed transcriptomic associations reflect functional biological mechanisms or correlative patterns. Second, pathway enrichment analysis remains inherently subjective due to arbitrary statistical thresholds for significance (e.g., FDR < 0.05 vs FDR < 0.1), which could lead to divergent biological interpretations depending on analytical parameters chosen by different researchers. Third, the retrospective design and reliance on TCGA data introduce inherent selection biases, necessitating validation in independent clinical cohorts with prospectively collected clinical outcomes. Fourth, while we identified candidate MPRGs associated with BCR, functional validation is required to establish causal relationships between these genes and prostate cancer progression. Finally, our nomogram model might benefit from inclusion of additional clinical variables such as PSA kinetics and radiographic findings to improve prognostic accuracy.

## Conclusion

In summary, we established a novel MPRG-derived signature to forecast the BCR in prostate cancer. Additionally, the association between MPRGs and the TIME, as well as chemotherapeutic response could offer potential guidance for personalized treatment strategies.

## Supporting information

**S1 Table. Motor protein-related genes obtained from the KEGG database.**
(XLSX)

**S2 Table. Primer sequences.**
(XLSX)

## Author contributions

**Conceptualization:** Weixing Wang, Guohai Xie, Zhonggao Wu.

**Data curation:** Weixing Wang, Guohai Xie, Zhonggao Wu.

**Formal analysis:** Weixing Wang, Guohai Xie, Zhonggao Wu.

**Funding acquisition:** Weixing Wang, Guohai Xie, Zhonggao Wu.

**Investigation:** Weixing Wang, Guohai Xie, Zhonggao Wu.

**Methodology:** Weixing Wang, Guohai Xie, Zhonggao Wu.

**Project administration:** Weixing Wang, Guohai Xie, Zhonggao Wu.

**Resources:** Weixing Wang, Guohai Xie, Zhonggao Wu.

**Software:** Weixing Wang, Guohai Xie, Zhonggao Wu.

**Supervision:** Weixing Wang, Guohai Xie, Zhonggao Wu.

**Validation:** Weixing Wang, Guohai Xie, Zhonggao Wu.

**Visualization:** Weixing Wang, Guohai Xie, Zhonggao Wu.

**Writing – original draft:** Weixing Wang, Guohai Xie, Zhonggao Wu.

**Writing – review & editing:** Weixing Wang, Guohai Xie, Zhonggao Wu.

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
