## [Decision Letter · Decision Letter 0]

25 Mar 2025

PONE-D-25-09189Development of a Prostate Cancer Biochemical Recurrence Risk Signature Using Machine Learning and Motor Protein-Related GenesPLOS ONE

Dear Dr. Wang,

Thank you for submitting your manuscript to PLOS ONE. After careful consideration, we feel that it has merit but does not fully meet PLOS ONE’s publication criteria as it currently stands. Therefore, we invite you to submit a revised version of the manuscript that addresses the points raised during the review process.

We look forward to receiving your revised manuscript.

Kind regards,

Elnaz Pashaei, Ph.D

Academic Editor

PLOS ONE

3. In the online submission form, you indicated that [The data that support the findings of this study are available from the corresponding author upon reasonable request].

4. Please include your tables as part of your main manuscript and remove the individual files. Please note that supplementary tables (should remain/ be uploaded) as separate "supporting information" files

Additional Editor Comments:

1.The model performs well in TCGA-PRAD (AUC = 0.966) but poorly in MSKCC2010 (AUC = 0.739–0.684), suggesting potential dataset bias. Please address this issue.

2.The nomogram is developed using TCGA-PRAD but lacks external validation, which is essential to assess its reliability in broader populations.

3.While pathway enrichment analysis is performed, the study does not discuss how these pathways relate to prostate cancer progression or treatment response.

Reviewers' comments:

Reviewer's Responses to Questions

**Comments to the Author**

1. Is the manuscript technically sound, and do the data support the conclusions?

Reviewer #1: Yes

Reviewer #2: Partly

2. Has the statistical analysis been performed appropriately and rigorously? 

Reviewer #1: Yes

Reviewer #2: Yes

3. Have the authors made all data underlying the findings in their manuscript fully available?

Reviewer #1: Yes

Reviewer #2: Yes

4. Is the manuscript presented in an intelligible fashion and written in standard English?

Reviewer #1: Yes

Reviewer #2: Yes

5. Review Comments to the Author

Reviewer #1: Nice in silico exploratory study of motor protein transcriptomic signature and its correlation to biochemical recurrence in prostate cancer. Please if you could add additional paragraph to discussion section where you state certain limitations of the study, mainly that it is in silico and that as in any in silico study depending where you cut off the statistical significance in pathways enrichment analysis you get a different biological conclusions.

Reviewer #2: Thanks for the chance to review your work. This paper aimed to construct a risk signature of biochemical recurrence for prostate cancer, by using the methods of machine learning and the data from TCGA, KEGG, and MSKCC2010. Overall, this paper is solid and provided valuable insights for the understanding of Motor proteins and PCa progression. I have some minor suggestions for authors to refer to.

1) There are so many figures included in the paper, and please consider to conprehensively rearrange and reduce the number of all the figures. I think some figures should be merged into one.

2) Please consider to add a table to summarize all the related studies.

3) Some sentences are confusing to follow, and authors should carefully check all the typos and grammar.

6. PLOS authors have the option to publish the peer review history of their article (what does this mean? ). If published, this will include your full peer review and any attached files.

**Do you want your identity to be public for this peer review?** For information about this choice, including consent withdrawal, please see our Privacy Policy .

Reviewer #1: **Yes: ** Benjamin Benzon

Reviewer #2: No

---

## [Author Response · Author response to Decision Letter 1]

12 May 2025

Response: We have revised the format of the title page and main text of the manuscript in accordance with the journal's requirements.

2.Please note that PLOS ONE has specific guidelines on code sharing for submissions in which author-generated code underpins the findings in the manuscript. In these cases, we expect all author-generated code to be made available without restrictions upon publication of the work. Please review our guidelines at https://journals.plos.org/plosone/s/materials-and-software-sharing#loc-sharing-code and ensure that your code is shared in a way that follows best practice and facilitates reproducibility and reuse.

RESPONSE: Our team is not willing to share the code of this study for the time being.

3. In the online submission form, you indicated that [The data that support the findings of this study are available from the corresponding author upon reasonable request].

Response: We have revised the Data Availability section at the end of the manuscript: The RNA expression profiles and corresponding clinical information used in this study are available in the TCGA (https://portal.gdc.cancer.gov), cBioPortal for Cancer Genomics (https://www.cbioportal.org/), and GEO (https://www.ncbi.nlm.nih.gov/geo/query/acc.cgi?acc=GSE70769) databases. All relevant data are within the manuscript and its Supporting Information files.

4.Please include your tables as part of your main manuscript and remove the individual files. Please note that supplementary tables (should remain/ be uploaded) as separate "supporting information" files

Response: We have inserted the Tables immediately after their first citation in the main text and uploaded the supplementary tables as separate "supporting information" files.

Additional Editor Comments:

1.The model performs well in TCGA-PRAD (AUC = 0.966) but poorly in MSKCC2010 (AUC = 0.739–0.684), suggesting potential dataset bias. Please address this issue.

Response: To address this issue, we first further added another cohort, GSE70769, which contains 92 prostate cancer patients and their follow-up information. We then used the sva package to remove batch effects among cohorts. After conducting machine learning analysis, we found that the problem mentioned by the reviewer still occurred based on the original optimal model selection criteria. Therefore, we adopted a new method: ranking the models according to the mean C-index in the validation cohort, and dividing the cohorts into high- and low-risk groups based on the median risk score. We selected models with a p-value less than 0.05 in the Log-Rank test between the high- and low-risk groups across all cohorts as the optimal models. Using this new selection method, we resolved the issue you raised and revised the subsequent results based on the new model.

2.The nomogram is developed using TCGA-PRAD but lacks external validation, which is essential to assess its reliability in broader populations.

Response: We further validated the reliability of the nomogram in the MSKCC2010 and GSE70769 cohorts. Using receiver operating characteristic (ROC) analysis, we calculated the AUC values for predicting 1-, 3-, and 5-year biochemical recurrence (BCR) in the above-mentioned cohorts.

3.While pathway enrichment analysis is performed, the study does not discuss how these pathways relate to prostate cancer progression or treatment response.

Response: We have thoroughly discussed the relationship between these pathways and prostate cancer progression or treatment response in the Discussion section of the revised manuscript.

Reviewer comments

Reviewer #1: Nice in silico exploratory study of motor protein transcriptomic signature and its correlation to biochemical recurrence in prostate cancer. Please if you could add additional paragraph to discussion section where you state certain limitations of the study, mainly that it is in silico and that as in any in silico study depending where you cut off the statistical significance in pathways enrichment analysis you get a different biological conclusions.

Response: We have addressed the limitations of the study in the final paragraph of the Discussion section as follows: While this in silico study provides novel insights into the prognostic value of MPRG expression signatures in prostate cancer, several important limitations must be acknowledged. First, the entirely computational nature of this investigation inherently limits biological interpretation, as we cannot confirm whether observed transcriptomic associations reflect functional biological mechanisms or correlative patterns. Second, pathway enrichment analysis remains inherently subjective due to arbitrary statistical thresholds for significance (e.g., FDR <0.05 vs FDR <0.1), which could lead to divergent biological interpretations depending on analytical parameters chosen by different researchers. Third, the retrospective design and reliance on TCGA data introduce inherent selection biases, necessitating validation in independent clinical cohorts with prospectively collected clinical outcomes. Fourth, while we identified candidate MPRGs associated with BCR, functional validation is required to establish causal relationships between these genes and prostate cancer progression. Finally, our nomogram model might benefit from inclusion of additional clinical variables such as PSA kinetics and radiographic findings to improve prognostic accuracy.

Reviewer #2: Thanks for the chance to review your work. This paper aimed to construct a risk signature of biochemical recurrence for prostate cancer, by using the methods of machine learning and the data from TCGA, KEGG, and MSKCC2010. Overall, this paper is solid and provided valuable insights for the understanding of Motor proteins and PCa progression. I have some minor suggestions for authors to refer to.

1) There are so many figures included in the paper, and please consider to conprehensively rearrange and reduce the number of all the figures. I think some figures should be merged into one.

Response: Based on the results of the re-analysis, we have combined Figures 2 and 3 to reduce the number of figures and have also revised some overlapping and unclear elements within the figures.

2) Please consider to add a table to summarize all the related studies.

Response: We have added a table (Table 2) in the Discussion section to summarize the studies on the related markers. Since there are many studies on the functions and mechanisms of these genes in cancer, we mainly summarized the functions and mechanisms of the eight MPRG marker genes in prostate cancer.

3) Some sentences are confusing to follow, and authors should carefully check all the typos and grammar.

Response: We have systematically reviewed the sentences throughout the manuscript to make them more comprehensible and have also corrected some spelling and grammatical errors. Specifically:

Changed "being" to "and is" on line 47.

Changed "role" to "importances" on line 19.

Added "thus" on line 58.

Added two commas on line 62.

Changed "could" to "may" on line 75.

Changed "along with" to "as well as" on line 286.

Changed "include" to "incorporates" on line 303.

Added "and" on line 360.

Added "and/" on line 368.

Added "for improving outcomes" on lines 369-370.

In addition, we have checked and proofread the sections modified due to other comments for grammatical and phrasing issues.

---

## [Decision Letter · Decision Letter 1]

8 Jul 2025

PONE-D-25-09189R1Development of a Prostate Cancer Biochemical Recurrence Risk Signature Using Machine Learning and Motor Protein-Related GenesPLOS ONE

Dear Dr. Wang,

Thank you for submitting your manuscript to PLOS ONE. After careful consideration, we feel that it has merit but does not fully meet PLOS ONE’s publication criteria as it currently stands. Therefore, we invite you to submit a revised version of the manuscript that addresses the points raised during the review process.

We look forward to receiving your revised manuscript.

Kind regards,

Elnaz Pashaei, Ph.D

Academic Editor

PLOS ONE

Journal Requirements:

Additional Editor Comments:

Please address my requests below in addition to the reviewer comments:

1) While external cohorts were included, the conclusions rely entirely on bioinformatics analyses. Experimental or clinical validation would substantially strengthen the findings. Please update your manuscript to include further validation of your findings, and ensure that cohort matching on clinical covariates is fully addressed. 

2) Although key clinical variables such as T and N stages were analyzed, important factors like PSA levels, treatment history, and comorbidities were not included or discussed. Please incorporate these to improve the robustness of the prognostic model.

Reviewers' comments:

Reviewer's Responses to Questions

**Comments to the Author**

1. If the authors have adequately addressed your comments raised in a previous round of review and you feel that this manuscript is now acceptable for publication, you may indicate that here to bypass the “Comments to the Author” section, enter your conflict of interest statement in the “Confidential to Editor” section, and submit your "Accept" recommendation.

Reviewer #3: (No Response)

Reviewer #4: All comments have been addressed

Reviewer #5: All comments have been addressed

Reviewer #6: (No Response)

2. Is the manuscript technically sound, and do the data support the conclusions?

Reviewer #3: Yes

Reviewer #4: Yes

Reviewer #5: Yes

Reviewer #6: Yes

3. Has the statistical analysis been performed appropriately and rigorously? 

Reviewer #3: Yes

Reviewer #4: Yes

Reviewer #5: Yes

Reviewer #6: Yes

4. Have the authors made all data underlying the findings in their manuscript fully available?

Reviewer #3: Yes

Reviewer #4: Yes

Reviewer #5: Yes

Reviewer #6: No

5. Is the manuscript presented in an intelligible fashion and written in standard English?

Reviewer #3: Yes

Reviewer #4: Yes

Reviewer #5: Yes

Reviewer #6: No

6. Review Comments to the Author

Reviewer #3: (No Response)

Reviewer #4: This second version of the paper is a great improvement, the authors are to be commended.

The manuscript has been much improved and is in a nice condition now.

Reviewer #5: All suggestions have been addressed. The statistical analysis has been conducted appropriately and rigorously and any biases have been highlighted.

Reviewer #6: The Manuscript presented by Weixing Wang et al entitled: "Development of a Prostate Cancer Biochemical Recurrence Risk Signature Using Machine Learning and Motor Protein-Related Genes" is technically sound and presents a comprehensive bioinformatics and machine learning pipeline to develop a prognostic model for biochemical recurrence (BCR) in prostate cancer based on motor protein-related genes (MPRGs).

1. Technical soundness:

The use of two independent cohorts (TCGA-PRAD and MSKCC2010) for training and validation strengthens the reliability of the findings. the study's conclusions are generally well-supported by the presented data, including differential gene expression, survival analysis, ROC curves, immune cell infiltration correlations, and drug sensitivity predictions. However, the biological interpretation of some genes within the 8-MPRG signature could benefit from more mechanistic explanation or references, especially for less characterized genes in prostate cancer context.

2. statistical analysis:

The statistical approach is appropriately rigorous, especially with the incorporation of 10 machine learning algorithms and 101 model combinations. The authors applied uninvariate Cox regression, leave-one-out cross-validation (LOOCV), and evaluated performance using Harrel's index and ROC/AUC analysis- all of which are acceptable and correctly used.

That said, some aspects could be strengthened:

- the model calibration could be described more clearly. Were the calibration curves statistically evaluated?

- Multivariate Cox regression was performed, but the interpretation that riskscore is the only independent predictor (Table 1) requires further context (e.g., confidence intervals, hazard ratios).

3. Manuscript presentation and language:

The manuscript is generally intelligible and logically structured, with standard scientific English. However, there are numerous grammatical and typographical issues that should be addresssed to improve clarity. Some sentences are overly long and awkwardly constructed.

Examples:

- Line 13: "but their involvement in the biochemical recurrence (BCR) of prostate cancer is not well understood" can be improved as " but their role in biochemical recurrence (BCR) of prostate cancer remains unclear."

- Line 183: Age was observed to be positively associated with the riskscores (r=0.16, p=0,0053, Figure 4D), indicating that older patients had significantly higher risk scores compared to younger patients (Figure 4E)." can be improved as " Age was positively correlated with risk scores (r=0.16, p=0.0053; Figure 4D), suggesting that older patients tend to have higher riskscores than younger individuals (Figure 4E)."

- Line 23: Consider revising "with 8 significantly associated with BCR" to "of which 8 were significantly associated with BCR."

- Figure Legends are repetitive; consider condensing for example:

Figure 3 legend: this legend restates what each panel obviously shows, without adding interpretive value or clarification. it's descriptive but nor insightful. it can be improved as " Performance of the MPRG-based RSF model in predicting BCR. Risk score distributions, Kaplan-Meier survival curves, and ROC analyses for 1-, 3-, and 5-year BCR prediction are shown for the TCGA-PRAD (A-C) and MSKCC2010 (D-F) cohorts. Please consider same condensing to other legends as well.

- Data availability:

Refer to (Line328-330): "The data that support the findings of this study are available from the corresponding author upon request." However, earlier in the form-based metadata (near the beginning of the PDF), the authors checked "Yes- all data are fully available without restriction." this is inconsistent and contradicts PLOS ONE's open data policy. As the data appear to be sourced from public databases (e.g., TCGA, cBioPortal), the authors should revise statement to include direct links or accession numbers to these sources and clearly indicate that no restrictions apply.

- Novelty and impact:

The manuscript offers techical novelty through its use of motor protein-related genes in a machine learning framework to predict biochemical recurrence in prostate cancer. while the computational framework id rigorous and the biological rationale is sound, the novelty would be strengthened by comparing this model against existing BCR signatures and discussing its added clinical value over current available tools. Moreover, the lack of experimental validation or prospective cohort analyses limits the immediate applicability of the model in clinical settings.

In summary, this manuscript presents a technically sound and biologically motivated approach to predicting biochemical recurrence in prostate cancer using machine learning and motor protein-related genes. while the study is methodologically rigorous and offers meaningful insights, it would benefit from improved language clarity, enhanced biological interpretation, and alignment with open data standards. I recommend minor to moderate revision before acceptance.

7. PLOS authors have the option to publish the peer review history of their article (what does this mean? ). If published, this will include your full peer review and any attached files.

**Do you want your identity to be public for this peer review?** For information about this choice, including consent withdrawal, please see our Privacy Policy .

Reviewer #3: **Yes: ** In my opinion, the article contains all possible ways of developing the topic and can be published in its original version.

Reviewer #4: No

Reviewer #5: No

Reviewer #6: **Yes: ** Shayan Maleknia

---

## [Author Response · Author response to Decision Letter 2]

13 Aug 2025

Dear editor and reviewers

Thank you for taking the time to review our manuscript titled "Development of a prostate cancer biochemical recurrence risk signature using machine learning and motor protein-related genes (Manuscript ID: PONE-D-25-09189R1)". We have carefully considered all of your feedback, and we are pleased to report that we have addressed each point raised in your review with the utmost attention to detail. Please note that the modifications in the revised draft are highlighted in red.

Journal Requirements:

Response: We have reviewed the references to ensure they meet the requirements.

Additional Editor Comments:

Please address my requests below in addition to the reviewer comments:

1)While external cohorts were included, the conclusions rely entirely on bioinformatics analyses. Experimental or clinical validation would substantially strengthen the findings. Please update your manuscript to include further validation of your findings, and ensure that cohort matching on clinical covariates is fully addressed.

Response: In this revision, we have supplemented the in vitro gene expression analysis, comparing the expression of MPRGs between PCa cell lines DU145 and LNCaP and the benign prostate hyperplasia epithelial cell line BPH-1, further validating the results of the bioinformatics analysis.

2)Although key clinical variables such as T and N stages were analyzed, important factors like PSA levels, treatment history, and comorbidities were not included or discussed. Please incorporate these to improve the robustness of the prognostic model.

Response: In this revision, we have included PSA and treatment history in the analysis. The results show that PSA is also an independent prognostic factor for prostate cancer. We have refined the nomogram and its performance evaluation results. Since the MSKCC cohort does not have PSA data, we have only validated the performance of the nomogram in the GEO cohort. Please refer to Table 1, Fig 7, and the section on Construction of Nomogram Using MPRG-Derived Risk Score for detailed results.

Reviewer #6: The Manuscript presented by Weixing Wang et al entitled: "Development of a Prostate Cancer Biochemical Recurrence Risk Signature Using Machine Learning and Motor Protein-Related Genes" is technically sound and presents a comprehensive bioinformatics and machine learning pipeline to develop a prognostic model for biochemical recurrence (BCR) in prostate cancer based on motor protein-related genes (MPRGs).

1. Technical soundness:

The use of two independent cohorts (TCGA-PRAD and MSKCC2010) for training and validation strengthens the reliability of the findings. the study's conclusions are generally well-supported by the presented data, including differential gene expression, survival analysis, ROC curves, immune cell infiltration correlations, and drug sensitivity predictions. However, the biological interpretation of some genes within the 8-MPRG signature could benefit from more mechanistic explanation or references, especially for less characterized genes in prostate cancer context.

Response: Thank you very much for your comments. However, you should be reviewing the initial version, as some of the key issues you mentioned have already been revised in the first revision. Regarding this issue, in the first revision, we have conducted an exhaustive literature analysis and organization of the 8 MPRGs, which have been compiled in Table 2.

2. statistical analysis:

The statistical approach is appropriately rigorous, especially with the incorporation of 10 machine learning algorithms and 101 model combinations. The authors applied uninvariate Cox regression, leave-one-out cross-validation (LOOCV), and evaluated performance using Harrel's index and ROC/AUC analysis- all of which are acceptable and correctly used.

That said, some aspects could be strengthened:

- the model calibration could be described more clearly. Were the calibration curves statistically evaluated?

- Multivariate Cox regression was performed, but the interpretation that riskscore is the only independent predictor (Table 1) requires further context (e.g., confidence intervals, hazard ratios).

Response: In our analysis, we used the rms::calibrate() function for calibration curve analysis, which is adopted by the majority of similar studies. This function does provide statistical evaluation in a Cox model (fitted by cph()), but these evaluations are indirect and quantitative, rather than a single p-value or χ² statistic like the Hosmer-Lemeshow (HL) test applicable to generalized linear models. It mainly assesses the model's calibration performance by generating a calibration object, focusing on correcting overfitting and quantifying prediction bias. It uses bootstrapping or cross-validation methods to estimate the model's optimism, generates calibration curves, and combines visual inspection for analysis. We have added these details to the Materials and Methods section. Additionally, we have further described the details of the calibration curve in the Results section. Moreover, we have re-analyzed the data according to the reviewers' and editors' comments, and the new results incorporating the risk score, T stage, and PSA data have been used to construct the nomogram. Therefore, the updated Cox regression results have also been revised.

3. Manuscript presentation and language:

The manuscript is generally intelligible and logically structured, with standard scientific English. However, there are numerous grammatical and typographical issues that should be addresssed to improve clarity. Some sentences are overly long and awkwardly constructed.

Examples:

- Line 13: "but their involvement in the biochemical recurrence (BCR) of prostate cancer is not well understood" can be improved as " but their role in biochemical recurrence (BCR) of prostate cancer remains unclear."

- Line 183: Age was observed to be positively associated with the riskscores (r=0.16, p=0,0053, Figure 4D), indicating that older patients had significantly higher risk scores compared to younger patients (Figure 4E)." can be improved as " Age was positively correlated with risk scores (r=0.16, p=0.0053; Figure 4D), suggesting that older patients tend to have higher riskscores than younger individuals (Figure 4E)."

- Line 23: Consider revising "with 8 significantly associated with BCR" to "of which 8 were significantly associated with BCR."

Response: In this revision, we have made the changes as per your suggestions, systematically checked and corrected spelling errors and grammatical mistakes in the manuscript, and improved some sentences that were too long and awkwardly constructed.

- Figure Legends are repetitive; consider condensing for example:

Figure 3 legend: this legend restates what each panel obviously shows, without adding interpretive value or clarification. it's descriptive but nor insightful. it can be improved as " Performance of the MPRG-based RSF model in predicting BCR. Risk score distributions, Kaplan-Meier survival curves, and ROC analyses for 1-, 3-, and 5-year BCR prediction are shown for the TCGA-PRAD (A-C) and MSKCC2010 (D-F) cohorts. Please consider same condensing to other legends as well.

Response: We have streamlined the figure legends as per your suggestion.

- Data availability:

Refer to (Line328-330): "The data that support the findings of this study are available from the corresponding author upon request." However, earlier in the form-based metadata (near the beginning of the PDF), the authors checked "Yes- all data are fully available without restriction." this is inconsistent and contradicts PLOS ONE's open data policy. As the data appear to be sourced from public databases (e.g., TCGA, cBioPortal), the authors should revise statement to include direct links or accession numbers to these sources and clearly indicate that no restrictions apply.

Response: We have revised the data availability paragraph and provided the link for data access.

- Novelty and impact:

The manuscript offers techical novelty through its use of motor protein-related genes in a machine learning framework to predict biochemical recurrence in prostate cancer. while the computational framework id rigorous and the biological rationale is sound, the novelty would be strengthened by comparing this model against existing BCR signatures and discussing its added clinical value over current available tools. Moreover, the lack of experimental validation or prospective cohort analyses limits the immediate applicability of the model in clinical settings.

Response: In this revision, we have conducted in vitro analysis of the gene expression of the 8 MPRGs to further validate the results of the bioinformatics analysis.

---

## [Decision Letter · Decision Letter 2]

4 Sep 2025

Development of a Prostate Cancer Biochemical Recurrence Risk Signature Using Machine Learning and Motor Protein-Related Genes

PONE-D-25-09189R2

Dear Dr. Wang,

We’re pleased to inform you that your manuscript has been judged scientifically suitable for publication and will be formally accepted for publication once it meets all outstanding technical requirements.

Kind regards,

Elnaz Pashaei, Ph.D

Academic Editor

PLOS ONE

Additional Editor Comments (optional):

Reviewer #6:

Reviewers' comments:

Reviewer's Responses to Questions

**Comments to the Author**

1. If the authors have adequately addressed your comments raised in a previous round of review and you feel that this manuscript is now acceptable for publication, you may indicate that here to bypass the “Comments to the Author” section, enter your conflict of interest statement in the “Confidential to Editor” section, and submit your "Accept" recommendation.

Reviewer #6: All comments have been addressed

2. Is the manuscript technically sound, and do the data support the conclusions?

Reviewer #6: Yes

3. Has the statistical analysis been performed appropriately and rigorously? 

Reviewer #6: Yes

4. Have the authors made all data underlying the findings in their manuscript fully available?

Reviewer #6: Yes

5. Is the manuscript presented in an intelligible fashion and written in standard English?

Reviewer #6: Yes

6. Review Comments to the Author

Reviewer #6: Dear Authors,

I would like to thank you for your careful revisions and thoughtful responses to all reviewer comments. the manuscript has beensubstantially improved in terms of clarity, language, and presentation. the statistical analysis are now much clear, the figures and legends are more concise, and the discussion better contextualizes the biological significance of the findings.

The revised data availability statement also now meets PLOS ONE's open-data requirements.

Overall, the study is technically sound, well presented, and makes a meaningful contribution by developing a machine learning-based signiture of motor protein-related genes for predicting biochemical recurrence in prostate cancer.I have no further concerns and support publication of this work.

Best regards,

7. PLOS authors have the option to publish the peer review history of their article (what does this mean? ). If published, this will include your full peer review and any attached files.

**Do you want your identity to be public for this peer review?** For information about this choice, including consent withdrawal, please see our Privacy Policy .

Reviewer #6: **Yes: ** Shayan Maleknia

---

## [Editor Report · Acceptance letter]

PONE-D-25-09189R2

PLOS ONE

Dear Dr. Wang,

I'm pleased to inform you that your manuscript has been deemed suitable for publication in PLOS ONE. Congratulations! Your manuscript is now being handed over to our production team.

Kind regards,

on behalf of

Dr. Elnaz Pashaei

Academic Editor

PLOS ONE